

# Improving ancient DNA genome assembly

Alexander Seitz and Kay Nieselt

Center for Bioinformatics (ZBIT), Integrative Transcriptomics, Eberhard-Karls-Universität Tübingen, Tübingen, Germany

## ABSTRACT

Most reconstruction methods for genomes of ancient origin that are used today require a closely related reference. In order to identify genomic rearrangements or the deletion of whole genes, *de novo* assembly has to be used. However, because of inherent problems with ancient DNA, its *de novo* assembly is highly complicated. In order to tackle the diversity in the length of the input reads, we propose a two-layer approach, where multiple assemblies are generated in the first layer, which are then combined in the second layer. We used this two-layer assembly to generate assemblies for two different ancient samples and compared the results to current *de novo* assembly approaches. We are able to improve the assembly with respect to the length of the contigs and can resolve more repetitive regions.

# INTRODUCTION

The introduction of next generation sequencing (NGS) made large scale sequencing projects feasible (*Bentley et al., 2008*). Their high throughput allows for fast and cheap sequencing of arbitrary genomic material. It revolutionized modern sequencing projects and made the study of ancient genomes possible (*Der Sarkissian et al., 2015*). However, the resulting short reads pose several challenges for the reconstruction of the desired genome when compared to the longer Sanger reads (*Li et al., 2010*; *Sawyer et al., 2012*). For modern DNA samples, the problem of having only short reads can be mitigated by the sheer volume of sequenced bases and usage of long fragments with paired-end and mate-pair sequencing. The insert size is used to determine the distance between the forward and the reverse read, which are sequenced from both ends of the fragments. These distances can be important for *de novo* assembly as they are used for repeat resolution and scaffolding. However, samples from ancient DNA (aDNA) mostly contain only very short fragments between 44 and 172 bp (*Sawyer et al., 2012*). Paired-end sequencing of these short fragments therefore often results in overlapping forward and reverse reads (thus actually negative inner mate pair distances). Because of these short fragments, mate-pair sequencing as well as sequencing technologies producing long reads (like PacBio) do not result in the same information gain that can be achieved on modern samples. Additionally, post-mortem damage of aDNA, most importantly the deamination of cytosine to uracil, can result in erroneous base incorporations (*Rasmussen et al., 2010*). Using reference based

Corresponding author
Alexander Seitz, alexander.seitz@uni-tuebingen.de

approaches, these errors can be detected, as they always occur at the end of the fragments. This is not possible using *de novo* assembly approaches and these errors can lead to mistakes in the assembly. However, treating the sample with *Uracil-DNA Glycosylase* (UDG) can resolve most of these errors (*Briggs et al., 2010*). Deeper sequencing does not always yield better results, because the amount of endogenous DNA contained in aDNA samples is often very low (*Sawyer et al., 2012*).

In order to achieve a higher content of endogenous DNA, samples of ancient origin are often subject to enrichment using capture methods (*Avila-Arcos et al., 2011*). The principle of these capture methods relies on selection by hybridization (*Maricic, Whitten & Pääbo, 2010*). Regions of interest are fixed to probes prior to sequencing. These probes can be immobilized on glass slides, called array capture (*Hodges et al., 2007*), or recovered by affinity using magnetic beads, referred to as in-solution capture (*Gnirke et al., 2009*). Using these capture methods, DNA fragments that can bind to the probes are used for amplification, which increases the amount of the desired DNA. However, as these methods amplify sequences that are contained on the probes, regions that were present in ancient samples and lost over time are not amplified and thus cannot be identified as they are not specifically targeted (*Khan, un Nabi & Iqbal, 2013*). Because of the low endogenous DNA content of the samples, many aDNA projects use these capture methods (*Shapiro & Hofreiter, 2014*).

In order to reconstruct a genome from sequencing data produced with next-generation technologies, one can either align the reads against a given, closely related reference genome or use so-called *de novo* assembly approaches, which are solely based on the sequencing information itself (*Nagarajan & Pop, 2010*; *Hofreiter et al., 2015*). In the former case, mapping programs like BWA (*Li & Durbin, 2009*) or Bowtie (*Langmead & Salzberg, 2012*) are popular methods that are especially suited for short reads. After the reads have been aligned, single nucleotide variations (SNVs) or short indels between the reconstructed genome sequence of the sample and the reference genome can be identified.

Because of the inherent characteristics of aDNA, specialized mapping pipelines for the reconstruction of aDNA genomes, such as EAGER (*Peltzer et al., 2016*) and PALEOMIX (*Schubert et al., 2014*), have recently been published. Mapping against a reference genome allows researchers to easily eliminate non-endogenous DNA and identify erroneous base incorporations. These errors can be identified after the mapping, e.g., by mapDamage (*Ginolhac et al., 2011*) or PMDtools (*Skoglund et al., 2014*), and used to verify that the sequenced fragments stem from ancient specimen.

The reference-based mapping approaches cannot detect large insertions or other genomic architectural rearrangements. In addition, if the ancient species contained regions that are no longer present in the modern reference, these cannot be identified by mapping against modern reference genomes. In these cases a *de novo* assembly of the genome should be attempted. This is also true for modern samples, if no closely related reference is available. The introduction of NGS has lead to new assembly programs, such as SOAPdenovo2 (*Luo et al., 2012*), SPADES (*Bankevich et al., 2012*), and many more that can handle short reads. However, if the ancient sample was sequenced after amplification through capture arrays, genomic regions that are not contained on the probes also can't be identified. Using shotgun

sequencing, reads originating from species that colonized the sample post-mortem are often more abundant (*Knapp & Hofreiter, 2010*). However, if shotgun data are available an effort for assembly can be made to identify longer deletions or genomic rearrangements.

The assembly of modern NGS data is still a challenging problem (*Chao, Yuan & Zhao, 2015*) and methods to improve it are still being developed. Among these is ALLPATHS-LG (*Gnerre et al., 2011*), arguably the winner of the so-called Assemblathon (*Earl et al., 2011*). ALLPATHS-LG uses the information provided by long fragments from paired-end and mate-pair sequencing to improve the assembly, and has therefore been shown to be one of the best assembly programs that are available today (*Utturkar et al., 2014*). However, because of the short fragments contained in aDNA samples, this approach is not feasible for aDNA projects and other methods have to be employed.

*De Bruijn* graph assemblers highly rely on the length of the $k$-mer to generate the graph (*Li et al., 2012*). The choice of an optimal value is even a difficult problem for modern sequencing projects (*Durai & Schulz, 2016*).

Because of the short fragments of aDNA samples, the sequencing adapter is often partially or fully sequenced (*Lindgreen, 2012*). After the adapter is removed, the length of the resulting read is then equal to the length of the fragment. Furthermore, overlapping forward and reverse reads can be merged to generate longer reads, which is usually done in aDNA studies to improve the sequence quality (*Peltzer et al., 2016*). Thus, the length distribution of reads from aDNA samples is often very skewed. This implies that the choice of one single fixed $k$-mer size in *de Bruijn* graph-based assembly approaches is not ideal in aDNA studies. Long $k$-mers miss all reads that are shorter than the value of $k$ and shorter $k$-mers cannot resolve repetitive regions.

In order to overcome the problem of the different input read lenghts, we have developed a two-layer assembly approach. In the first layer, the contigs are assembled from short reads using a *de Bruijn* graph approach with multiple $k$-mers. These contigs are then used in the second layer in order to combine overlapping contigs contained in the different assemblies resulting from the first layer. This is done using an overlap-based approach.

The next section contains the methods we used to improve and compare the *de novo* assembly for aDNA samples. In 'Results', we used our two-layer assembly to improve the assembly of two ancient DNA samples and compare our approach to different assembly programs.

## METHODS

The general structure of our two-layer assembly approach is to use multiple assemblies in a first layer with different $k$-mers, which are then merged in a second layer assembly using an overlap-based assembly program (see Fig. 1).

We used the tool *Clip & Merge* (*Peltzer et al., 2016*) to preprocess the reads. In order to evaluate how different preprocessing affects the assembly, the reads were all adapter clipped, quality trimmed, and then treated using three different methods: First, *Clip & Merge* was used with default parameters to merge overlapping forward and reverse reads. Second, the parameter `-no_merging` was used to perform only adapter-clipping and

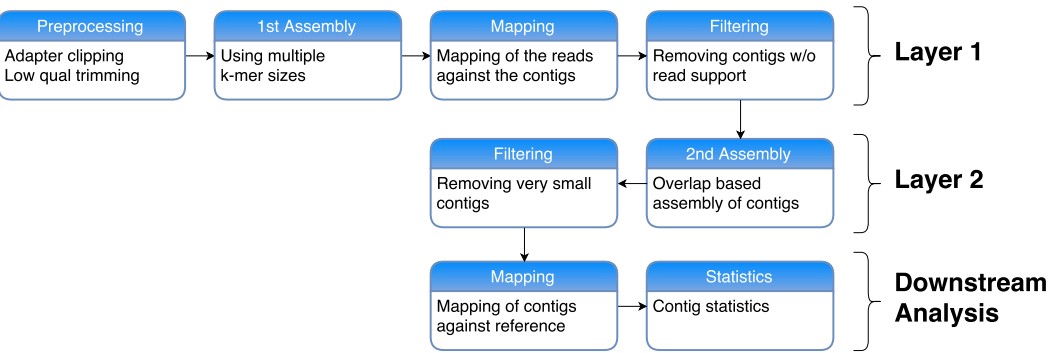

**Figure 1** **Workflow of our two-layer assembly approach.** First the reads are preprocessed by removing sequenced adapters and clipping low-quality bases. After that, multiple *de novo* assemblies are generated using a *de Bruijn* graph approach with multiple values for *k*. The reads are then mapped back against each of these resulting contigs and the contigs with no read support are filtered out. In Layer 2, these filtered contigs are then combined and assembled again using an Overlap-Layout-Consensus approach. Very short contigs are removed. The resulting contigs are mapped against a reference genome and contig statistics are calculated in order to assess the quality of the assembly.

quality-trimming without the merging of the reads, leaving the paired-end information (reads with no partner were removed). Third, after processing the reads as described in the second method, we gave each read a unique identifier and combined all forward and reverse reads in one file. Here reads without partners were kept. After the first and third method, a single-end assembly was performed, whereas the reads from the second preprocessing method were used in a paired-end assembly.

The different preprocessing methods result in reads of different length. The reason for this are the different fragment lengths contained in the sample. To resolve problems originating from these different lengths, we propose assembly of aDNA using a two-layer approach. In the first layer, we use a *k*-mer based assembly program. For our analysis here, we used SOAPdenovo2 (*Luo et al., 2012*) and MEGAHIT (*Li et al., 2014*) in the first layer, but any other assembly program, for which different values for *k* can be chosen, can be used. In order to cover a broad range of *k*-mers representing both short and long reads contained in the input, we used ten different *k*-mer sizes $(37, 47, 57, \ldots, 127)$.

*De Bruijn* based programs first generate all possible *k*-mers based on the input reads. Matching *k*-mers are used to generate the *de Bruijn* graph. This can lead to random overlaps of *k*-mers contained in different reads and therefore to read incoherent contigs (*Myers, 2005*). To filter out the contigs generated by random overlaps, we used BWA-MEM (*Li, 2013*) to map the reads against contigs. Contigs that are not supported by any read were removed before the next step.

To merge the results of the different assemblies of the first layer, each contig is given a unique identifier before they are combined into one file. This file is the input of the second layer assembly. Here, the assembly is based on string overlaps instead of *k*-mers, a concept originally introduced by *Myers (2005)*. An assembly program that uses this approach is the String Graph Assembler (SGA) (*Simpson & Durbin, 2012*). It efficiently calculates all overlaps of the input using suffix arrays (*Manber & Myers, 1993*). These overlaps are then

used to generate an overlap graph and the final contigs are generated based on this graph. We used this method to merge the contigs from the different assemblies based on their overlap.

As SGA uses string-based overlaps and modern sequencing techniques are not error-free, it provides steps to correct for these errors. There is a preprocessing step that removes all bases that are not A,G,C or T. There is also a correction step that performs a $k$-mer based error correction and a filtering step that removes input reads with a low $k$-mer frequency. Because the input for SGA are pre-assembled contigs, these errors should already be averaged out (*Schatz, Delcher & Salzberg, 2010*) and these steps were not used for the assembly of the second layer. However, the assemblies with the different $k$-mers produce similar contigs, which is why the duplicate removal step of SGA is performed. SGA can also use the Ferragina Manzini (FM) index (*Ferragina & Manzini, 2000*) to merge unambiguously overlapping sequences, which was used to further remove duplicate information. Afterwards, the overlap graph was calculated and the new contigs were assembled. All these steps were performed using the standard parameters provided by SGA. Afterwards, contigs shorter than 1,000 bp were removed from the final assembly. In order to evaluate our two-layer assembly method, the resulting contigs were then aligned with the reference genome of interest. We used again BWA-MEM for this step. Finally various statistics for the assembly were computed.

The results are compared to other *de Bruijn* assembly programs that can use information from multiple $k$-mer sizes to generate their assembly graph. Both SOAPdenovo2 and MEGAHIT can use the information from several $k$-mers, which is why we also evaluate against these results. Additionally, we use the "interactive *de Bruijn* graph de novo assembler" (IDBA) (*Peng et al., 2010*), in order to get results from an assembly program that was not part of our two-layer assembly evaluation and also uses multiple $k$-mers for the generation of the assembly graph. To evaluate the results using only an overlap-based approach, we also assembled the preprocessed input reads directly with SGA.

In order to evaluate our two-layer assembly approach, we applied it to two different published ancient samples. One is the sample Jorgen625, published by *Schuenemann et al. (2013)* containing DNA from ancient *Mycobacterium leprae*, the other one is the sample OBS137, published by *Bos et al. (2016)* containing DNA from ancient *Yersinia pestis*. There are two sequencing libraries available for the sample Jorgen625. In order to evaluate the two leprosy libraries as well as the OBS137 sample, we used the EAGER pipeline (*Peltzer et al., 2016*) to map the libraries against the respective reference genome (*Mycobacterium leprae TN* and *Yersinia pestis CO92*).

## RESULTS

The application of EAGER to the two *Mycobacterium leprae* libraries of Jorgen625 revealed that one of them contained relatively long fragments with a mean fragment length of 173.5 bp and achieved an average coverage on the reference genome of 102.6X. The other library was sequenced on an Illumina MiSeq with a read length of 151 bp. It was produced from shorter fragments with a mean fragment length of 88.1 bp and a mean coverage of

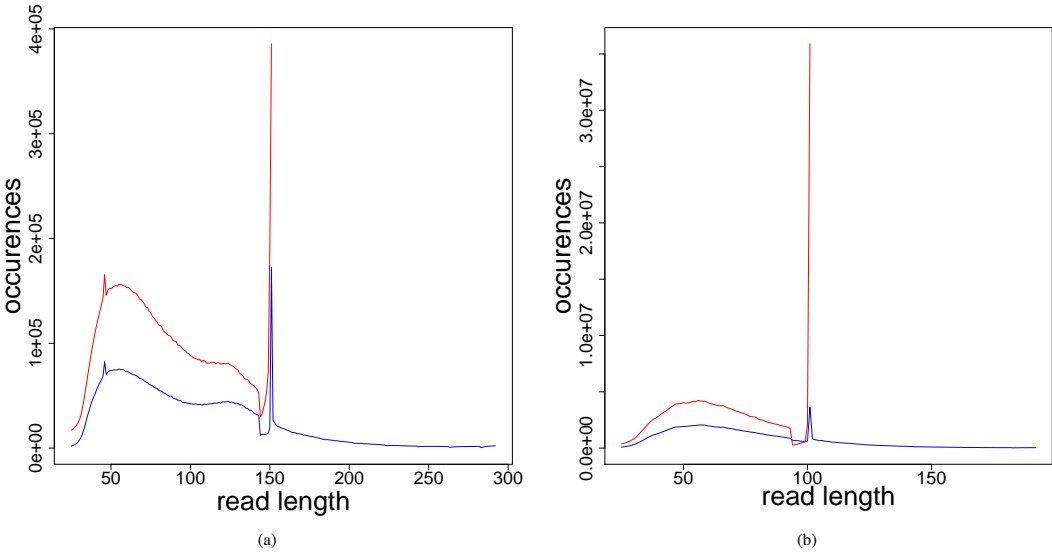

**Figure 2** **Read length distribution for the different preprocessed (adapter clipped and quality trimmed) `fastq` files for (A) Jorgen625 (*Mycobacterium leprae*) and (B) OBS137 (*Yersinia pestis*).** Red, unmerged reads; blue, reads after merging.

49.3X. With its shorter fragments and lower achieved coverage, the second library better reflects typical sequencing libraries generated from aDNA samples (*Sawyer et al., 2012*), so we focused our experiments on this library. The OBS137 sample was sequenced on an Illumina HiSeq 2000 with a read length of 101 bp. The mean fragment length of this library is 69.2 bp and achieved a mean coverage of 279.5X. It is important to note that the leprosy data were generated using shotgun sequencing, whereas the pestis data was first amplified using array capture methods. Both samples were treated with UDG.

The distribution of the read lengths after the preprocessing steps (see Fig. 2) shows that the resulting read lengths are highly variable. The peak at read length 151 (in the leprosy case) and 101 (in the pestis case), respectively, are attributed to those reads that were sequenced from fragments longer than the read length. For these no adapter and no low quality bases had to be removed. Therefore, after preprocessing they have the original read length performed in the respective experiment.

For the comparison of the different assembly programs, we extracted the contigs that can be mapped against the respective reference genome (*Mycobacterium leprae TN* and *Yersinia pestis OBS137*, resp.) and calculated several statistics (see Table 1). The results that were generated in the second layer are shown as well as the assembly that generated the longest contig in the first layer using the respective assembly program. Additionally, results from SGA applied to the reads themselves as well as results from programs that can use multiple $k$-mers in their assembly are shown. The complete result table with all intermediate steps is available in Supplemental Information 1.

For both samples, the values for the longest contig, the N50, and the mean contig length could be almost doubled by our two-layer approach. On the leprosy sample, the best result was achieved using all clipped input reads in one single-end assembly without merging. On

**Table 1  Results using our two-layer assembly with SOAPdenovo2 and MEGAHIT compared to the separate assemblies of SGA, SOAPdenovo2, MEGAHIT and IDBA.** The results show only values for contigs that could be mapped against the respective reference genome. Only the best assemblies (w.r.t. the longest mapped contig) for the different preprocessing methods and $k$-mers are shown. "SOAP" represents the results using multiple $k$-mers for the generation of their graph structure. "MEGAHIT" and "IDBA" alone also represent an assembly using multiple internal $k$-mers. The assemblies next to "Lyr X" represent the best assemblies generated by our approach in Layer X = 1 or 2. Preprocessing refers to how the reads were preprocessed before assembly and gaps represent the number of gaps that result after the contigs were mapped against the reference genome. Values in bold represent the top value of the respective metric that were be achieved per sample (see first column). All other statistical values can be found in Supplemental Information 1.

| | | Name | Prepro-cessing | # contigs ≥1,000 bp | N50 | Mean contig length | Longest contig | # gaps |
|---|---|---|---|---|---|---|---|---|
| *Mycobacterium leprae* | Separate | SOAP | Single | 249 | 21,909 | 13210.3 | 99866 | 103 |
| | | MEGAHIT | Merged | 175 | 28,410 | 16777.5 | 91499 | 106 |
| | | IDBA | Paired | 164 | 35,419 | 20152.7 | 118220 | 118 |
| | | SGA | Single | 1,157 | 2,199 | 1997.3 | 8640 | 952 |
| | Lyr 1 | SOAP K57 | Single | 215 | 24,962 | 14918.6 | 72345 | 120 |
| | | MEGAHIT K77 | Merged | 253 | 21,863 | 12765.4 | 87880 | 108 |
| | Lyr 2 | SOAP + SGA | Single | **133** | **42,136** | **25225.0** | **135656** | 88 |
| | | MEGAHIT + SGA | Merged | 668 | 19,758 | 12245.3 | 109259 | **80** |
| *Yersinia pestis* | Separate | SOAP | Single | 1,745 | 2,263 | 2098.9 | 8641 | 1,034 |
| | | MEGAHIT | Merged | 1,090 | 4,042 | 3267.1 | 9972 | 640 |
| | | IDBA | Merged | **779** | **5,196** | **3839.1** | 9988 | 498 |
| | | SGA | Merged | 3 | 1,126 | 1291.7 | 1633 | 6 |
| | Lyr 1 | SOAP K47 | Merged | 91,112 | 131 | 118.3 | 6425 | 901 |
| | | MEGAHIT K77 | Single | 4,940 | 1,321 | 898.6 | 6307 | 1,980 |
| | Lyr 2 | SOAP + SGA | Merged | 1,960 | 2,633 | 2281.0 | **13420** | 842 |
| | | MEGAHIT + SGA | Single | 3,104 | 1,884 | 1816.7 | 11478 | 967 |

the pestis sample, the best result was achieved using the merged input reads. Using both SOAPdenovo2 and MEGAHIT with multiple $k$-mers for the generation of the assembly graph, the overall assembly was improved by up to 30% compared to the single $k$-mer assembly. Using SGA directly on the preprocessed reads did not result in good assembly results when compared to SOPA, MEGAHIT or IDBA. IDBA produced the best results when compared to any other assembly using only one layer. On the pestis data, it also overall produced the best results except when comparing the length of the longest contig. Here the longest contig produced by our two-layer approach was up to 35% longer than the one computed by IDBA. For the leprosy data, all statistical metrics were lower compared to our two-layer assembly.

The length distribution of the resulting leprosy contigs shows a clear shift towards longer contigs (see Fig. 3). Because the contigs generated from the pestis data were very short, we did not filter them for a minimum length of 1,000 bp. It can be seen that even when all contigs are used, there is a shift towards longer contigs after our two-layer assembly method.

Since one normally is interested in one genome of interest, we computed the genome coverage after mapping all contigs of length at least 1,000 bases against the reference genome of *Mycobacterium leprae TN*. We used Qualimap2 (*Okonechnikov, Conesa &*

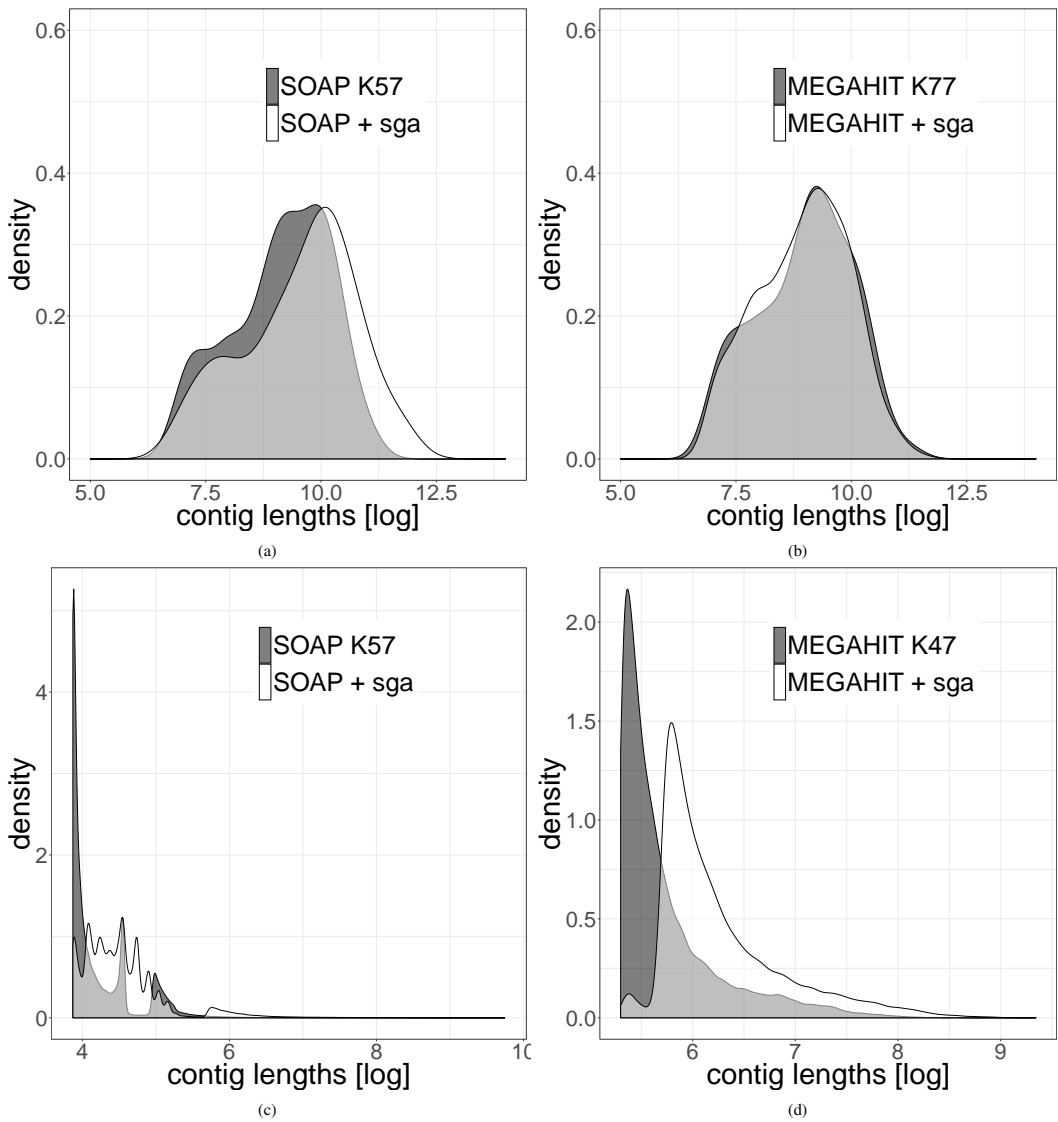

**Figure 3** **Distribution of the length of the contigs generated by the different assemblies.** The results generated by the second layer assembly with SGA are shown in white. The results of one first layer assembly is shown in dark grey. The light grey part represents the overlap of both methods. (A) Shows the results using SOAPdenovo2 in the first layer and (B) shows the results using MEGAHIT in this layer for the leprosy data. (C and D) Show the same results on the pestis data. In order to highlight the differences, the data were logarithmized.

*García-Alcalde, 2015*) for the analysis of the mapping. We also analyzed the coverage of the leprosy genome, that could be achieved using only contigs longer than 1,000, 1,500, ..., 10,000 bp (see Fig. 4). It shows that the percentage of the genome that could be covered is always higher after the second layer assembly than using only the results generated in the first layer assemblies. This becomes more and more pronounced with increasing filter threshold for the minimum contig length. The drop in coverage that results from the removal of shorter contigs is lower for our two-layer approach than using only first-layer assemblies. When using only contigs longer than 1,000 bp, the results are almost the same.

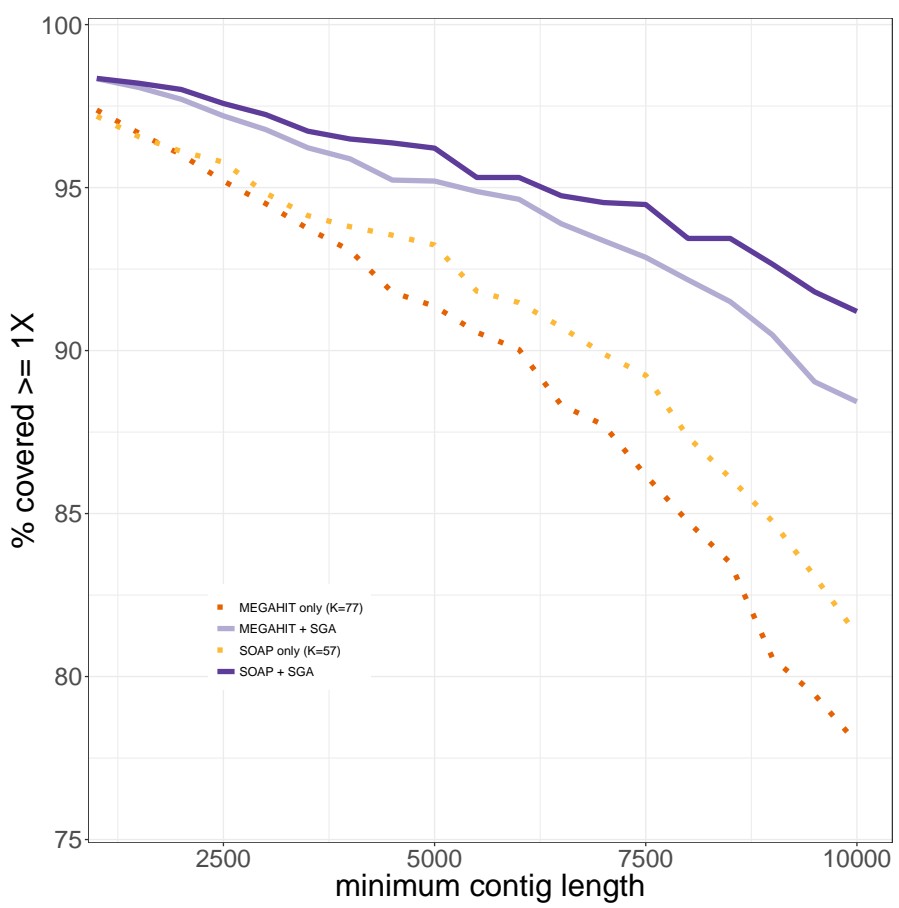

**Figure 4** **The percentage of the reference genome of *Mycobacterium leprae TN* that could be covered using only contigs longer than the minimum contig length.** Results from the first and second layer assemblies are shown.

Using only contigs longer than 10,000 bp, around 90% of the genome can be covered using the second layer assembly with SGA, whereas at most 80% of the genome is covered by contigs from assemblies generated in the first layer. This means that the same percentage of coverage of the reference genome can be achieved with longer contigs in comparison to the results generated in the first layer. When filtering the pestis data for contigs with a minimum length of 1,000 bp, the best coverage by assemblies of the first layer that could be achieved was 60%. The coverages that could be achieved by the second layer assemblies range between 70 and 83%, where each assembly improved on the ones of the first layer by at least 16% (see Supplemental Information 1). Analyzing the mapped contigs that were generated by the second layer, we found that they mapped almost perfectly (with some small insertions and deletions) against the reference genome.

The percentage of the genome that was covered more than once is around 1% for the assemblies generated in the first layer with SOAPdenovo2 and MEGAHIT. This value has increased after the second layer assembly where the contigs were assembled again with SGA, showing that not all overlapping contigs could be identified and merged by SGA.
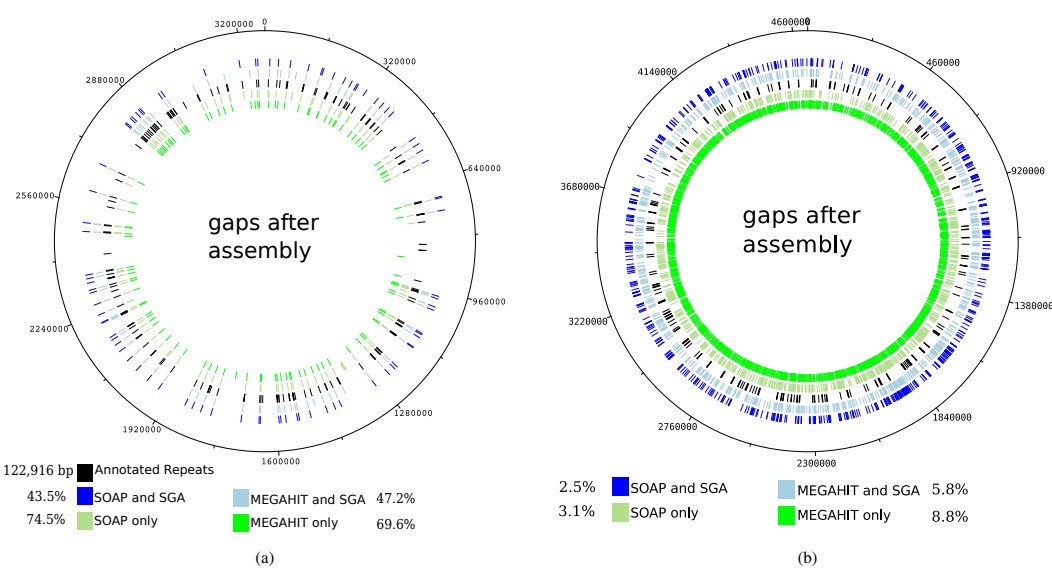

**Figure 5** **Gaps in the mapping of the contigs against the reference genome of *Mycobacterium leprae TN* (A) and *Yersinia pestis CO 92* (B) together with annotated repeat regions in the respective reference genome.** The outer ring represents the gaps that occur after the mapping of the contigs that were generated by the second layer assembly with SGA after a first layer assembly with SOAPdenovo2. The second outer ring shows the same but for a first layer assembly using MEGAHIT. The middle ring represents the annotated repeat regions of the reference genome. The second inner and innermost ring represent the gaps after using the best individual SOAPdenovo2 and MEGAHIT assemblies, respectively. The percentages represent the relative number of unresolved bases in annotated repeat regions for *Mycobacterium leprae* (in total 122,916 bp). For *Yersinia pestis*, the respective values represent the percentage of unresolved genomic positions.

The mapping of the contigs generated by the first layer assemblies of SOAPdenovo2 and MEGAHIT against the reference genome of *Mycobacterium leprae TN* resulted in 108 and 120 gaps, depending on the assembly program (see Table 1). These values were reduced to 80 and 88 gaps, respectively, for the contigs generated by the second layer assembly with SGA. It can be seen that for the leprosy genome, the gaps in the mapping of the contigs mainly coincide with annotated repeat regions in the reference genome, as already shown by *Schuenemann et al. (2013)* (see Fig. 5A). Altogether, the percentage of unresolved repetitive regions has dropped from 74.5% (when using only SOAPdenovo2) down to 43.5% using our two-layer approach.

For the pestis genome, this is not the case, as the resolved regions to not coincide with repetitive regions. However, it is apparent that after our two-layer approach, more genomic regions could be resolved. When analyzing the mapping of the raw reads against the reference genome of *Mycobacterium leprae TN* with Qualimap2 (*Okonechnikov, Conesa & García-Alcalde, 2015*), 100% of the genome could be covered at least once and 99–98% of the genome was covered at least five times.

Up until now we showed that we were able to generate long, high quality contigs that can be mapped against the respective reference. Because the leprosy data were generated from shotgun sequencing, we analyzed whether the assembled contigs actually belong to the species of *Mycobacterium leprae* and not to other *Mycobacteria*. For this we took the

ten longest contigs from each assembly and used BLASTN (*Altschul et al., 1990*), available on the NCBI webserver, to align the contigs with all the genomes available from the genus *Mycobacterium*. All hits that generated the highest score for all of these 10 contigs belonged to a strain of *Mycobacterium leprae* (data not shown). As the pestis data were generated using a capture approach and *Yersinia pestis* typically cannot survive longer than 72 h in soil (*Eisen et al., 2008*), the contamination of other *Yersinia* bacteria can be excluded, which is why we did not perform this experiment on the pestis data.

Furthermore, we evaluated the scalability of our pipeline through subsampling. We used the library from the Jorgen625 sample with the longer fragments, as it contained more than twice as many reads ($2 \times 15{,}101{,}591$ instead of $2 \times 6{,}751{,}711$ reads). We evaluated the whole pipeline using 1, 2, 5, 10 and all 15.1 million reads. The calculations were performed on a server with 500 GB available memory and 32 CPUs of type Intel® XEON® E5-416 v2 with 2.30 GHz. We evaluated the pipeline using four threads wherever parallelization was possible. The results show that the runtime scales linearly with the number of input reads (see Fig. S1). The time it would take to assemble a human genome using our two-layer approach can be estimated using a linear regression. The ancient human LBK/Stuttgart sample published by *Lazaridis et al. (2014)* was sequenced using eight lanes, each containing between 200 and 230 million reads. The assembly of one such lane would take approximately one week and the assembly of all 1.74 billion reads almost two months.

## DISCUSSION AND CONCLUSIONS

It has been shown that *de novo* genome assembly quality depends on sequencing coverage, read accuracy, and read length (*Nagarajan & Pop, 2013*; *Myers Jr, 2016*). With ancient genome assembly one faces a number of additional challenges. The underlying dataset stems from a metagenomic sample with short fragments. When performing a paired-end sequencing experiment, this results in mostly overlapping forward and reverse reads. Because of the highly different read lengths after the necessary preprocessing steps, including adapter removal and quality trimming, typical *de Bruijn* approaches using a fixed $k$-mer size cannot sufficiently assemble the sample. On the other hand, overlap-based approaches alone are also inferior. Our two-layer approach combining various assemblies using different $k$-mer sizes followed by a second assembly based on string overlaps is able to fuse the contigs generated in the first layer into longer contigs and reduce the redundancy. Additionally, we could show that longer, high quality contigs are generated after the second layer assembly. In particular, at least for our example genomes, we are able to resolve more gaps. In the example of the *Mycobacterium leprae* genome, these gaps mainly span repetitive regions. The different values for $k$ that are used in the first layer assembly lead to similar contigs that can be combined in the second layer assembly. The percentage of the genome that is covered more than once is increased after the second layer assembly of the leprosy data (see Supplemental Information 1). This shows that SGA is not able to identify and merge all overlapping contigs. One reason for this could be the underlying metagenomic sample combined with the shotgun sequencing approach. Multiple species

in the sample share similar but not identical sequences. As SGA is not designed to assemble metagenomic samples, these differences cannot be distinguished from different sequences of the same genome containing small errors. This theory is supported by the fact that on the pestis data, which were enriched using a capture array, this additional coverage was reduced but not eliminated in comparison to the first layers (see Supplemental Information 1). This signifies that when assembling metagenomic and especially aDNA samples, the results always have to be regarded critically to avoid mistakes. In order to identify contigs belonging to our desired genome, we mapped them against a closely related reference genome. The contigs that are generated after the second layer map almost perfectly against the reference sequence that is known to be highly similar to the desired genome (*Mendum et al., 2014*), showing that even though we are assembling a metagenomic sample, the generated contigs of interest are highly specific. However, because of the metagenomic sample, contigs of other species are also present in the assembly and have to be excluded.

Another possibility could be sequencing errors in the sample, leading to distinct contigs using different $k$-mers. However, these errors can be excluded as a possible source of error, as they should be averaged out by the different assemblies (*Schatz, Delcher & Salzberg, 2010*). Erroneous base incorporations are unlikely to be the source of these distinct contigs, as the sample was treated with *Uracil-DNA Glycosylase* (UDG), removing these errors. However, UDG does not repair methylated sites, so there may still be errors at sites of cytosine methylation (*Briggs et al., 2010*). Because the assemblies in the first layer are based on the majority of a base call at each position, given a high enough coverage (*Schatz, Delcher & Salzberg, 2010*), these errors should also be accounted for.

An important step is the preprocessing of the raw reads. We compared the performance using all reads as single reads, as paired reads or as merged reads. However, at least from our study, we can conclude that the results highly depend on the first layer assembler and probably also on the dataset itself. Interestingly, on the leprosy sample, SOAPdenovo2 produces better results when using all input reads in a single-end assembly than in a paired-end assembly. One possible explanation is that the information between the pairs does not contain additional information as almost all paired-end reads overlap and can be merged. It is possible that the program then disregards some overlaps in order to fulfill the paired-end condition. Overlaps that were disregarded this way could be used in the single-end assembly leading to a better assembly. Additionally, reads that did not have a partner were removed before the paired-end assembly. These reads are available in the single-end assembly. It could be that they contained some relevant information. On the pestis sample, the best results were achieved using the merged data. The reason for this is probably the length of the sequenced reads. In order to stay comparable, we used the same settings for the pestis data as for the leprosy data. However, because the pestis sample was sequenced with 101 bp reads, *de Bruijn* graph assemblers using a longer $k$-mer size than 101 bp cannot assembly anything. This means that the assemblies in the first layer using a $k$-mer size of 107, 117, and 127 could not produce any results. This does not hold true for the merged data, because the merging of the reads resulted in longer reads (up to 192 bp). Because of these longer input reads, these three assemblies contributed information that could then be used in the second layer assembly.

The mapping of the assembled contigs from the leprosy dataset against the reference show that in our case, all gaps align with annotated repeat regions (for the assembly using SOAPdenovo2 in the first layer). Using our two-layer assembly approach, more of these regions could be resolved, but many still remain. In sequencing projects of modern DNA, repetitive regions are resolved using other sequencing technologies such as PacBio. It can produce much longer sequences that span these regions. However, these technologies are not applicable to aDNA as most of the fragments contained in the sample are even shorter than the sequences that can be produced using the Illumina platforms.

In general, it can be concluded that assembly of aDNA is highly dependent on the amount of endogenous DNA in the sample and thus the coverage of each base (*Zerbino & Birney, 2008*). We are able to improve results generated by current assembly programs. However, the information gain generated by the second layer assembly is dependent on the quality of the first layer assemblies. Thus if the first layer assemblies are of low quality, the second layer assembly cannot improve them significantly. In the example of the pestis data, the second layer assembly could improve on the contigs generated in the first layer assemblies but could not create an almost perfect assembly, as was the case on the leprosy dataset where the contigs in the first layer assemblies were already of high quality. First tests showed that in order to achieve an assembly covering all but the repetitive regions continuously, the input reads should achieve at least a coverage of 10–15X, where more than 90% of the genome should be covered more than five times. Of course this is not the only criteria, which can be seen from the pestis data, so more experiments have to be done in order to identify the reasons that make the assembly of an ancient genome possible.

The runtime scales linearly with the number of input reads, which is no problem for small bacterial datasets. Since parallelization of our pipeline is straightforward, assembly of ancient human genome samples will also be feasible.

We have shown that our approach is able to improve the assembly of ancient DNA samples. However, this approach is not limited to ancient samples. In the paper by *Arora et al. (2016)*, we used this two-layer assembly approach on modern, hard to cultivate *Treponema pallidum* samples. The processing of these samples also resulted in only short fragments similar to ancient DNA. There, we were able to use our assembly approach to verify that a gene is missing in certain samples.

## SOFTWARE AVAILABILITY

We have developed an automated software pipeline, written in JAVA which will allow other researchers to use our methodology. This pipeline is available on GitHub: https://github.com/Integrative-Transcriptomics/MADAM.

## ACKNOWLEDGEMENTS

We thank Linus Backert and Vladimir Piven for their critical assessment and discussions on different ideas, as well as implementation specific details. Additionally we would to thank Alexander Peltzer, André Hennig, Michael Römer and Niklas Heinsohn for valuable

insights and discussions on different ideas, implementation specific questions and for their comments on this paper. Finally we want to thank Dr. Mathew Divine for his feedback regarding this paper.

### Funding

We received support from Deutsche Forschungsgemeinschaft and the Open Access Publishing Fund of University of Tübingen. The funders had no role in study design, data collection and analysis, decision to publish, or preparation of the manuscript.

### Grant Disclosures

The following grant information was disclosed by the authors:
Deutsche Forschungsgemeinschaft.
University of Tübingen.

### Competing Interests

Kay Nieselt is an Academic Editor for PeerJ.

### Author Contributions

- Alexander Seitz conceived and designed the experiments, performed the experiments, analyzed the data, wrote the paper, prepared figures and/or tables, reviewed drafts of the paper.
- Kay Nieselt wrote the paper, prepared figures and/or tables, reviewed drafts of the paper.

### DNA Deposition

The following information was supplied regarding the deposition of DNA sequences:
NCBI accession number SRS418854.

### Data Availability

GitHub: https://github.com/Integrative-Transcriptomics/MADAM.

### Supplemental Information

Supplemental information for this article can be found online at http://dx.doi.org/10.7717/peerj.3126#supplemental-information.

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
