# Peer review of "Improving ancient DNA genome assembly"

_PeerJ, doi:10.7717/peerj.3126_

## Round 0.1 · original submission · Major Revisions

In addition to the prior reviewing process from the German Conference on Bioinformatics I have asked an expert from the field of ancient DNA bioinformatics to review the manuscript. The referees' comments were generally positive and, in principle, we would be pleased to publish the manuscript. However, I ask that you consider the points raised by the referee. Please either answer them in a revised version or explain why they are not relevant.

·

Basic reporting

Basic reporting meets the standards and templates of this journal. However, I found serious issues with the presentation of the data and the language. I go into specific detail with an annotated manuscript. But in general the clarity of the text needs to be improved. I found a lot of repetition. The authors should review the English again a few times perhaps with the aid of a more fluent researcher.

I have reservations regarding the presentation and structure of the manuscript. I found that in sections the paragraph order and and placement could be improved. The paragraphs with the most important points are often in the middle of a section. More importantly the sections are not systematically ordered. With elements of the introduction in the M&M, M&M in results, results in discussion and elements that ought to be in the introduction found in the discussion.

Most importantly the article is desperately under referenced. Especially in the discussion which makes many sweeping statements without reference to back them up. In the introduction there needs to be more reference to technical papers on ancient DNA. Which are abundant.

Figures need to be tidied up. In general the axes appear messy. They are poorly referenced to in text too.

Experimental design

The design of the experiment is encouraging but lacks sufficient empirical data. I am confused as to why, with the great availability of aDNA metagenomes from a vast array of different studies, the authors chose to analyse such a small cohort. Worryingly they chose not to include the results of lower coverage data that were analysed but excluded. This needs to be addressed more substantially than the authors have done already. Otherwise the cohort size should be increased.

The efficacy of their approach is based on increased percentage of >1X mapping across the genome between several de novo assemblies. However, the authors acknowledge that these are in mainly non-specific repetitive regions that could come from several strains of mycobacteria or organisms. The authors need to discuss more substantially the possibility that the contigs may be assembled from several organisms and quantify in some way contamination.

Validity of the findings

Conclusions need to put to the end of the manuscript.

There needs to be more data analysed, using the vast amount of metagenomic data already available.

There needs to be more discussion as to the utility of filling in the gaps in the reconstructed genome. What information can be potentially gleaned? What new useful information does the assembled and re-mapped mycobacteria leprae genome have?

External reviews were received for this submission. These reviews were used by the Editor when they made their decision, and can be downloaded below.

---

## Round 0.2 · Minor Revisions

Thanks for addressing the issues raised in the previous review. Before we can accept the manuscript, it requires several minor changes and corrections. Please follow the recommendations of the reviewer, as indicated in the annotated PDF. We look forward to receiving your final version of the manuscript soon and promise to quickly process it.

·

Basic reporting

The basic reporting style is well done. However, I would recommend that the authors once again review the English language with the assistance of a more fluent speaker. The structure of the introduction could be revised, with the deletion of some redundant points. Also some points of information lack references to back them up (see attached review).

Experimental design

Graphs and table are a little unclear. The authors should revise their structure. Also data from NCBI search must be added to the supplement (see attached review).

Validity of the findings

no comment

Comments for the author

The points made are mostly minor in regards to presentation and structure. If these are resolved then I see no further reason not to publish this article.

---

## Round 0.3 · accepted · Accept

The concerns of the reviewers have all been satisfactorily addressed. Thanks!